

# The influence of fish farm activity on the social structure of the common bottlenose dolphin in Sardinia (Italy)

Serena Frau[1], Fabio Ronchetti[1], Francesco Perretti[1], Alberto Addis[1], Giulia Ceccherelli[2] and Gabriella La Manna[1,3]

[1] Environmental Research and Conservation, MareTerra Onlus, Alghero, Italy, Italy
[2] Dipartimento di Chimica e Farmacia, University of Sassari, Sassari, Italy, Italy
[3] Area Marina Protetta Capo Caccia Isola Piana, Alghero, Italy, Italy

## ABSTRACT

In a wide variety of habitats, including some heavily urbanised areas, the adaptability of populations of common bottlenose dolphin (*Tursiops truncatus*) may depend on the social structure dynamics. Nonetheless, the way in which these adaptations take place is still poorly understood. In the present study we applied photo-identification techniques to investigate the social structure of the common bottlenose dolphin population inhabiting the Gulf of Alghero (Sardinia, Italy), analysing data recorded from 2008 to 2019. The social structure analysis showed a division of the entire population into five different communities and the presence of non-random associations, while there was no evidence of segregation between sexes. Furthermore, results highlighted an important change in social structure through time, likely due to a reduction in fish farm activity since 2015. The division of the population into different communities, the presence of segregation based on the foraging strategy (inside or outside the fish farm area) and the social network measures were evaluated by analysing independently the two datasets: the intense and low farm activity periods: 2008–2014 and 2015–2020, respectively. Segregation among individuals belonging to the same foraging strategy class was found only in the earlier period, and the composition of the four communities was consistent with this result. Our study improves the knowledge about bottlenose dolphin adaptation, as a lower complexity in social structure was linked to a reduction in anthropogenic food availability.

## INTRODUCTION

Social organisation is the result of an evolutionary process and leads to the best balance between the benefits and costs of group living (*Alexander, 1974*; *Lehmann, Korstjens & Dunbar, 2007*). These costs and benefits can vary widely with the habitat and species-specific requirements; nevertheless, sexual segregation, food availability and predation risk are considered the most important drivers in shaping social systems (*Janson, 2000*; *Connor, Heithaus & Barre, 2001*; *Lehmann, Korstjens & Dunbar, 2007*; *Snaith & Chapman, 2007*). Living in large groups is beneficial in terms of group defences, but an increase in group size

Corresponding author
Gabriella La Manna,
mareterra.onlus@gmail.com

may also lead to stronger within-group competition (*Pulliam, 1973*; *Elgar, 1989*; *Roberts, 1996*; *Lingle & Wilson, 2001*).

An efficient strategy to overcome the costs of group living is to change the group size and composition over space and time: a social organisation called fission–fusion society (*Lehmann, Korstjens & Dunbar, 2007*). Among mammals, social systems characterised by fission–fusion dynamics are typical in some dolphins (*Connor et al., 2000*), but also some primates, elephants (*Wittemyer, Douglas-Hamilton & Getz, 2005*), spotted hyenas (*Holekamp et al., 1997*), some bats, and humans (*Rodseth et al., 1991*). In fission–fusion societies, individuals can modify the persistence of associations with others, making group composition a dynamic property (*Couzin, 2006*). Interactions between individuals are based on intrinsic factors such as the presence of preferred or avoided associates and extrinsic habitat characteristics, for example, prey density and habitat complexity (*Lusseau et al., 2003*). Fission–fusion societies can be investigated at different time scales. Over the apparent fluidity of this social organisation, it is possible to find strong and long-lasting relationships based on many factors, such as kinship, habitat utilisation and ecological constraints (*Couzin, 2006*).

Understanding animal social structure and interactions is pivotal for species conservation, since social structure can affect fitness (*Silk, 2007*), genetic structure (*Archie et al., 2008*) and the transmission of diseases and information (*Altizer et al., 2003*). The social structure of the common bottlenose dolphin (*Tursiops truncatus*, hereafter 'bottlenose dolphin') has been studied across different environments and locations (*Lusseau et al., 2003*; *Foley et al., 2010*; *Moreno & Acevedo-Gutierrez, 2016*; *Dinis et al., 2018*; *Pleslić et al., 2019*) and this species is generally considered the most adaptable among Odontocetes; by changing the diet and using innovative specialisations it can spread into a wide variety of habitats (*Reynolds, Wells & Eide, 2000*). For this reason, bottlenose dolphins are considered general and opportunistic feeders (*Bearzi, Fortuna & Reeves, 2009*; *Bearzi, Piwetz & Reeves, 2019*). Adaptation of the feeding behaviour to specific natural or human-related prey availability may also shape the social structure of populations (e.g., *Van Schaik & Van Hooff, 1983*).

One of the most investigated types of adaptation to human activity is the foraging around fishing gear and fish farms (*Chilvers & Corkeron, 2001*; *Díaz López & Shirai, 2008*; *Ansmann et al., 2012*; *Pace, Pulcini & Triossi, 2012*; *Genov et al., 2019*). Studies on the effect of fish farms on the social structure and distribution of common bottlenose dolphins have reported contrasting results. Around fish farms dolphins may be attracted by the high concentrations of wild prey (*Tuya et al., 2006*), likely due to the input of nutrients to feed the fish (*Fernandez-Jover et al., 2007*), and the structure of the cages themselves may facilitate prey capture (*Díaz López & Methion, 2017*), although some studies have revealed an avoidance of fish farming areas (*Markowitz et al., 2004*; *Díaz López, Polo & Marini, 2005*). In the Mediterranean Sea the bottlenose dolphin seem attracted by fish farms as a food source (*Díaz López & Shirai, 2008*; *Piroddi, Bearzi & Christensen, 2011*; *Díaz López, 2012*; *Pace, Pulcini & Triossi, 2012*; *Bonizzoni et al., 2014*; *Bonizzoni et al., 2019*). To our knowledge, it is currently unknown whether the social structure of the bottlenose dolphin population can be affected by a change in fish farm activity. However,

*Ansmann et al. (2012)* have related the social structure of the bottlenose dolphin to the change in fishing regime and suggested that the social structure was remodulated following a halving of prawn trawling activity. This change consisted of an increased connection among all individuals and a lack of differences between classes, likely due to the disappearance of different foraging strategies related to the presence of trawlers (*Ansmann et al., 2012*).

Thus, the aim of this study was two-fold. First, we investigated the social structure of the common bottlenose dolphin population inhabiting the Gulf of Alghero (Sardinia–western Mediterranean Sea), applying social structure analysis on 12-year photo-identification data. In particular, we assessed the presence of preferred and avoided associations between individuals, the presence of sex segregation, and clustering in different communities. Secondly, because the activity regime of the fish farm at the study site changed in the middle of the study period, we tested the hypothesis that the association pattern and social network structure of the dolphins changed following the variation in fish farm activity. Thus, we compared the level of associations between individuals, the social network measures and the clustering into communities in the two periods, i.e., the full and the reduced fish farm activity periods.

## MATERIALS & METHODS

### Study area and data collection

The study area (Fig. 1) is located in northwestern Sardinia (Italy) and extends for about 300 km$^2$ along the coast of Alghero (40.5580°N, 8.3193°E). It includes three different protected areas and a variety of different seafloor types: coastal gradually sloping rocky bottom, seagrass meadows (*Posidonia oceanica*) and detrital bottom exceeding 115 m in depth. The study was carried out under the permission of the Marine Protected Area Capo Caccia–Isola Piana (N° 0021343/2018). A seabass (*Dicentrarchus labrax*) and seabream (*Sparus aurata*) farm of six cages has been active in the study area since 2008. Due to a change in management, after 2015 the functioning of the fish farm has been discontinuous (as periods of activity alternated with periods of inactivity) and the number of cages was progressively reduced until 2018 when only two cages remained.

Data on bottlenose dolphins were collected from 2008 to 2019. Surveys were concentrated mainly in spring, summer and autumn due to the generally adverse weather conditions in winter. Systematic boat surveys were performed during daylight (from 9 am to 7 pm) by a 5.10-m research vessel powered by a 40-hp outboard engine and a 9.7-m motorboat powered by a 270-hp inboard engine, in only good sea and weather conditions (sea conditions <3 on the Douglas scale; wind force <3 on the Beaufort scale) and visibility of over 3 nm. Between 2008 and 2011, navigation routes followed linear transects, while from 2012 they were planned haphazardly (*La Manna, Ronchetti & Sarà, 2016*; *La Manna et al., 2020*); in both cases the whole study area was covered homogeneously and in almost every survey the fish farm area was visited. During navigation a mean boat speed of 8–18 km/h was maintained. In the proximity of the fish farm area (around 500 m) the boat speed was reduced to 4–8 km/h. The entire perimeter was then checked to verify the presence of dolphins around or inside the buoys delimiting the fish farm area. In the case of a dolphin

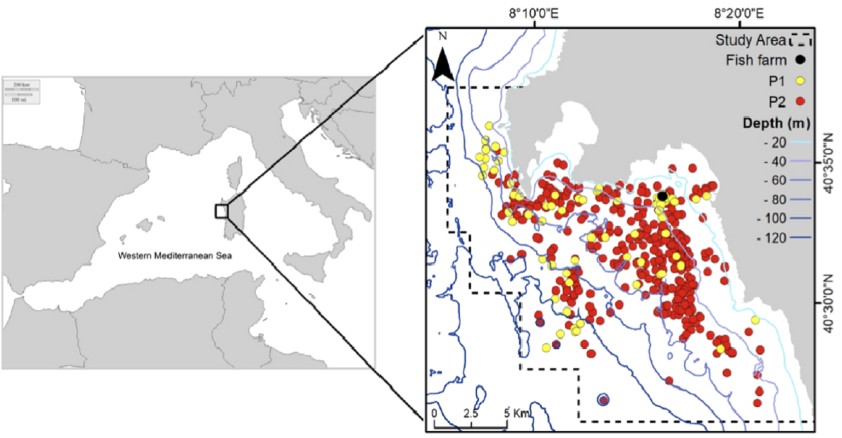

**Figure 1** **Study area and sightings considered in the analysis: yellow points (Period 1 dataset), red points (Period 2 dataset).** Black point: fish farm.

sighting, the animals were approached cautiously to avoid any disturbance. A sighting was defined as an observation of one or a group of dolphins. A group was defined as all individuals within visual range that were in apparent association, either by being close to each other and often, but not always, engaged in the same activity, or moving in the same direction (*Shane, 1990*). The location of each sighting was recorded using a GPS plotter.

During each sighting, data on group size and composition were recorded. Group size was determined *in situ* by two independent observers. Group composition was based on the classification of each individual in one of the following age classes (defined according to the relative body size): (1) adult: large and robust animals approximately 2.5–3 m long; (2) juveniles: animals about two-thirds adult size and slightly lighter in colour than an adult; (3) calves: animals about one-half adult size and usually swimming in association with the mother; (4) newborns: animals less than one month old, usually below one-half adult size and in constant physical contact with the mother (*Bearzi, Notarbartolo-Di-Sciara & Politi, 1997*). The sex of the individuals was determined by photographs depicting the genital area; if not available, individuals were classified as 'Not Determined' (ND). Moreover, individuals were sexed as females when repeatedly sighted in close relationship with a calf and presumed to be the mothers (*Shane, 1990*; *Mann, 2000*).

## Photo-identification and data organisation
Photo-identification allows the unique identification of individuals, using natural marks (nicks and notches) present on their body (*Würsig & Jefferson, 1990*). During each sighting, observers took photographs of the left and right side of the dorsal fin of each individual., using Nikon D90, D40 or D7000 cameras equipped with a 70–300 mm telephoto zoom lens, or a SONY Alpha 65 camera with 18–250 mm lens. Pictures were classified on the basis of their quality and the severity grade of the marks (*Urian et al., 2014*). High-quality images (based on focus, sharpness, exposure and angle) of well-marked individuals (based on the presence of clearly distinguishable nicks and notches) were compared with a photo-identification catalogue of known individuals. The catalogue was created using the

Darwin software and contains images and information about all the dolphins sighted at least once in the area during the study period.

Based on the photo-identification data, a dataset containing the composition of the groups, the number of calves and the percentage of the photo-identified individuals at each sighting was built. Then, a supplemental dataset was created with the age class (adult, juvenile, calf) and the sex (male, female, ND) of each individual. Furthermore, any individual and associated data was assigned to one or both of the following periods: (i) the full-activity period (P1) from 2008 to 2014, when the six-cage fish farm was continuously active, or (ii) the reduced activity period (P2) from 2015 to 2019, when the two-cage fish farm was intermittently active. We also distinguished individuals on the basis of their foraging strategy with respect to the fish farm area as follows: (i) individuals seen at least once foraging inside the perimeter of the fish farm area (defined as 'cage', C); (ii) individuals never seen inside the perimeter of the fish farm area (defined as 'non-cage', NC), where 'perimeter' is the limit of the area containing the cages, bordered by buoys and floating ropes. Although we did not make direct observation of prey-chasing behaviour inside the fish farm area, according to *Vermeulen, Holsbeek & Das (2015)*, we consider 'foraging' to be any surfacing patterns characterised by long dives (*dos Santos & Lacerda, 1987*; *Bearzi, Politi & Notarbartolo Sciara, 1999*) with frequent direction changes (*Rako-Gospić et al., 2020*). Moreover, since bottlenose dolphins inside the fish farm area have never been observed in other behavioural states, such as socialising, resting or escaping from predators, we consider our assumption of foraging as reasonable. Foraging behaviour of the group inside the fish farm was recorded using continuous focal follow group sampling (*Altmann, 1974*; *Mann, 2000*; *La Manna et al., 2019*; *La Manna et al., 2020*). For each individual we calculated the sighting rate: the number of sightings of the individual divided the total number of sightings. The sizes of the groups sighted while foraging inside the fish farm area were compared to those of the groups sighted outside the fish farm area by a Mann–Whitney U test (in fact the assumption of normality was not respected).

## Social associations of the entire population

Studies on animal social structure are often based on the rate of association between identified individuals. We calculated the coefficients of association using the half-weight index (HWI; *Cairns & Schwager, 1987*). HWI ranges between zero and one. A zero value indicates that the pair of individuals are never observed together in the same group; a score of 1 indicates that the pair of individuals are always observed in the same group. HWI was chosen because is the most relevant method when individuals are likely to be present in groups but not always identifiable (*Smolker et al., 1992*), and it allows comparisons with other studies (*Blasi & Boitani, 2014*; *Genov et al., 2019*), since it is the most-used association index.

For any individual observed several times in the same day, only the first sighting was counted, to avoid the non-independence and serial autocorrelation of sightings. Only individuals sighted at least five times in the study area were included in the analysis, to reduce bias in the measure of the association index. Calves were excluded from the analysis because of their strong association to the mother. Mean, maximum and minimum HWIs

were calculated for each individual, and an association matrix containing the association indices of each pair of individuals was created. To investigate whether or not individual HWIs were significantly different from random we used a Monte Carlo permutation test, randomly permuting associations within sampling periods according to *Bejder, Fletcher & Bräger (1998)*, with the modification suggested by *Whitehead, Bejder & Ottensmeyer (2005)*. We used the day as the sampling period. This method accounts for demographical effects such as birth, death or migrations that may occur on a longer time scale (*Whitehead, 2008a*). The number of permutations within daily sampling periods was increased up to the stabilisation of the *p-value* (10,000 permutations; *Bejder, Fletcher & Bräger, 1998*). Since the *p-value* cannot be considered as a statistical threshold to identify significant associations (*Whitehead, 2008b*), an arbitrary threshold was fixed to identify the significant associations at twice the mean association index of the population, including zero values (*Gero et al., 2005*). The coefficient of variation (CV) of the HWIs was used to test for long-term preferences (*Whitehead, 2008b*).

A Mantel test with 1,000 permutations was carried out to investigate the correlation between the association matrices between sexes. We excluded individuals of unknown sex from the analysis of sex segregation. The compiled version of SOCPROG 2.9 was used to compute HWI, the permutation and the Mantel tests.

The presence of different communities was detected using a Hierarchical Cluster analysis with the average linkage method, and the dendrogram was considered a good representation of the real division of the society if the value of the cophenetic correlation coefficient was ≥0.8 (*Whitehead, 2008b*). Community division was also investigated using Newman's eigenvector method and modularity 1 for gregariousness. A modularity greater than 0.3 indicates a significant division of the population (*Newman, 2004*). Cluster analysis and social division by modularity were carried out using the compiled version of SOCPROG 2.9. Graphical representation of the social structure (sociogram) was created using NetDraw (included in UCINet; *Borgatti, Everett & Freeman, 2002*). The associations between dyads with an association index twice the mean of the population, including zero values, were considered the preferred associations and were plotted in the sociogram (*Gero et al., 2005*). Further, we calculated the following network statistics: (1) the strength (the sum of association indices of any individual with all other individuals); (2) the reach (a measure of indirect connectedness of each individual); (3) the clustering coefficient (a measure of how the associates of an individual are themselves associated); and (4) the affinity (individual's neighbours weighted mean of the strength; *Whitehead, 2008b*).

## Effect of the fish farm: association indices, network metrics and analysis

To evaluate the effect of the fish farm activity over time the original dataset was split in two: one referring to as P1 (2008–2014) and the other P2 (2015–2019). Only individuals sighted at least five times in each period were included. This restriction resulted in P1 and P2 datasets containing 32 and 69 individuals, respectively. For each dataset, following the methods previously described, we calculated the HWI for each individual and constructed an association matrix containing the association indices of each pair of individuals. A

Mantel test was run to assess differences in association index between different foraging strategy classes (C vs NC). The null hypothesis was that associations between classes were similar. We also detected the presence of different communities by using Hierarchical Cluster analysis with average linkage and community division using Newman's eigenvector method and modularity 1 from gregariousness. The social structure of each period was represented by a sociogram. Further, we calculated network statistics for each dataset and assessed differences in the structures of social networks between foraging strategy classes (C vs NC) in each period and between P1 and P2, running two-sample permutation tests for each network measure using the software R (*R Development Core Team, 2015*).

# RESULTS

## General results

Overall, a total of 474 sightings were recorded during 596 surveys between September 2008 and October 2019 (overall 13,081 nm of survey effort). Specifically, in P1 during 205 surveys and 4,276 nm of effort 205 groups were sighted, while in P2 during 391 surveys and 8,099 nm of effort 269 groups were sighted. Since the fish farm is located in front of Alghero harbour, where each survey started and ended, the fish farm area was surveyed regularly (almost every survey), both in P1 and P2. Eighty-five individuals (over a total of 123 photo-identified dolphins) were included in the original dataset based on the condition that each one was seen at least five times between 2008 and 2019. Among them, 41 individuals were females, 13 males and 32 of unknown sex. The mean number of sightings per individual was 22.05 (SD = 18.15, range 4–87), while the mean sighting rate was 0.05 (SD = 0.04). According to the number of sightings for each individual, we identified four classes of residency (*Blasi & Boitani, 2014*): very frequent (40–87 times; 14% of individuals), frequent (20–39 times; 28% of individuals), low frequent (10–19 times; 34% of individuals) and rare (5–9 times; 24% of individuals). Bottlenose dolphins were sighted foraging inside the fish farm area 48 times (10% of the sightings): 25 individuals were classified as C ('cage') while the remaining 60 were assigned to the class NC ('no cage').

Mean group size, calculated for the pooled dataset, was 5.23 (SD = 3.97). Group size of those foraging inside the fish farm area was significantly lower (mean = 3.35, SD = 2.25) than that of the groups outside the fish farm area (mean = 5.45, SD = 4.07, Mann–Whitney U test: $p < 0.001$). Calves were present in 53% of the groups and only 2% were spotted foraging inside the fish farm area.

## Social analysis of the entire population

Pooled HWIs were calculated for each dyad ($n = 7225$, mean HWI ± SD = 0.05 ± 0.02; max HWI ± SD = 0.36 ± 0.13). Estimate of social differentiation using likelihood estimator gave a value of 1.007 (SE = 0.016), which indicates a well differentiated population (*Whitehead, 2008b*). The correlation between the true and estimated association indices was 0.619 (SE = 0.020), a reasonably good representation of the social system (*Ansmann et al., 2012*) since it is between $r = 0.4$ and $r = 0.8$: 'somewhat representative' and 'a good representation' of the real social system, respectively

(*Whitehead, 2008b*). Significantly higher SD (real SD = 0.08222, random SD = 0.08073, $p$ < 0.0001) and CV values in real data compared to random (real CV = 1.62799, random CV = 1.60185, $p < 0.0001$) revealed the presence of non-random associations in the population (*Whitehead & Default, 1999*). On average, each individual was preferentially associated with 16.92 individuals (SD = 7.32, range 2–32) out of the 85. Forty-two individuals had a higher than average number of associations (>17) and only two individuals were preferentially associated with only two others.

The mean HWI for females and males was 0.06 (SD = 0.03) and 0.05 (SD = 0.02), respectively. HWI within and between classes were similar, as the Mantel test run between males and females did not reveal differences in associations within and between sexes ($t = 0.525$, $p = 0.5994$). Thus, there was no apparent evidence of sex segregation within the population. The strength of associations in females were highly variable: (i) five females had no or just one significant association; (ii) twenty-two had significant associations with a low number of females (between 2 and 9); and (iii) thirteen had significant strong and long-lasting associations with numerous other females (from 10 to 18). Males in the Gulf of Alghero seemed to build only short-term and weak relationships compared to the female–female associations (max HWI ± SD = 0.33 ± 0.13). Two males never associated significantly with other males, two male individuals had associations with only one male each, while the other males showed significant associations with several males (from 3 to 6).

The cophenetic correlation coefficient (CCC) was not high enough (CCC = 0.760) to ensure a good representation of the association matrix, and the sociogram did not show a net distinction between communities (Fig. 2). Nonetheless, the maximum modularity (MM) of the population with division method based on modularity 1 (*Newman, 2004*) suggested a division of the society into five different communities (MM = 0.319). Communities 1, 2 and 4 had similar Mean and Max HWIs, similarly to that observed in the other communities (Table 1). Communities 3 and 5 formed stronger associations compared to the other communities. Community 5 was composed mainly of females, but the high number of ND individuals in this and all other communities did not allow us to understand their sex composition. Only Community 3 was composed entirely of C individuals, while the others were composed mainly of NC individuals. The two communities with the highest number of C individuals were Communities 2 and 3, while Communities 3 and 5 had the lowest number of total individuals (Table 2).

Regarding network measures for the entire population, the strength of associations was better correlated with affinity (correlation coefficient = 0.5588 SE = 0.0702) than with clustering coefficient (correlation coefficient = 0.1535, SE = 0.1072; Table 3), indicating that individuals associated preferentially with individuals with a similar number of associates (*Whitehead, 2008b*).

## Influence of the fish farm on social structure

The P1 dataset included 32 individuals: 20 C and 12 NC; 15 females, 5 males and 12 of unknown sex. Among these 32 individuals, 21 were also sighted in P2 (11 C and 10 NC). The percentage of times each C individual was sighted inside the fish farm area, calculated
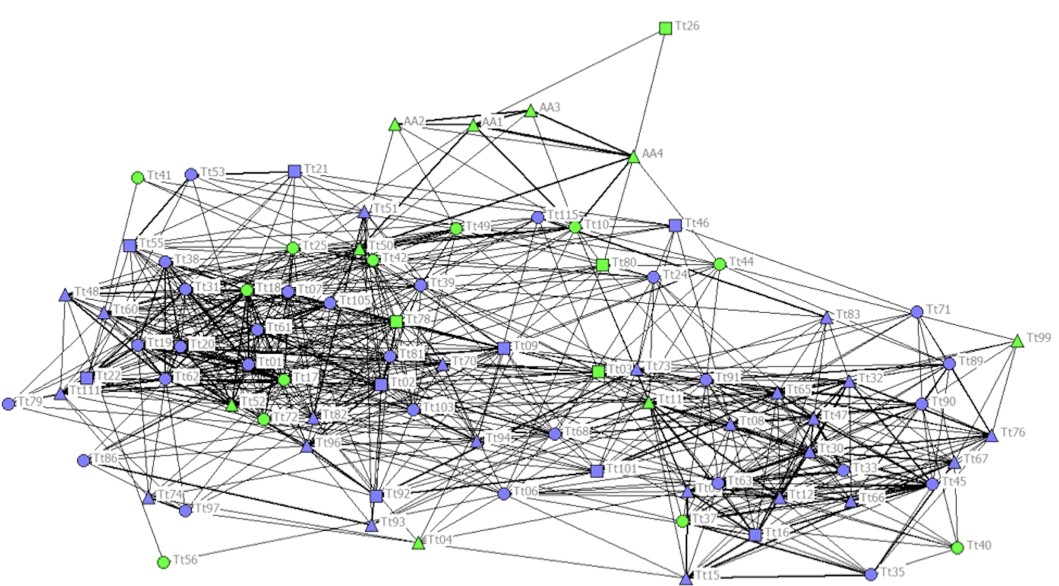

**Figure 2** Social network representation of the entire population, showing only the significant associations from the permutation test. Individuals are identified by their ID code; thickness of lines is proportional to HWI values. Nodes shapes represent different sexes (triangle = ND, circle = females, square = males). Green nodes are cage individuals (C), blue nodes are no cage individuals (NC).

**Table 1** Association within and between communities. Mean and Max HWI (±SD) of the five Communities detected by the maximum modularity (MM) of the population with division method based on modularity 1.

|  | Community1 | Community2 | Community3 | Community4 | Community5 |
|---|---|---|---|---|---|
| | | | **MEAN HWI (SD)** | | |
| Community1 | 0.11 (0.04) | 0.01 (0.02) | 0.01 (0.01) | 0.03 (0.03) | 0.01 (0.02) |
| Community2 | 0.01 (0.01) | 0.10 (0.04) | 0.03 (0.06) | 0.03 (0.02) | 0.07 (0.04) |
| Community3 | 0.01 (0.01) | 0.03 (0.02) | 0.25 (0.12) | 0.01 (0.02) | 0.01 (0.02) |
| Community4 | 0.03 (0.02) | 0.03 (0.02) | 0.01 (0.02) | 0.10 (0.03) | 0.07 (0.04) |
| Community5 | 0.01 (0.01) | 0.07 (0.04) | 0.01 (0.01) | 0.07 (0.03) | 0.23 (0.08) |
| | | | **MAX HWI (SD)** | | |
| Community1 | 0.36 (0.13) | 0.08 (0.07) | 0.03 (0.04) | 0.13 (0.09) | 0.07 (0.04) |
| Community2 | 0.13 (0.07) | 0.29 (0.11) | 0.08 (0.10) | 0.15 (0.05) | 0.17 (0.08) |
| Community3 | 0.05 (0.06) | 0.25 (0.13) | 0.48 (0.27) | 0.08 (0.10) | 0.05 (0.04) |
| Community4 | 0.18 (0.10) | 0.12 (0.05) | 0.07 (0.07) | 0.29 (0.06) | 0.17 (0.08) |
| Community5 | 0.09 (0.03) | 0.20 (0.08) | 0.06 (0.04) | 0.19 (0.06) | 0.42 (0.13) |

as the number of sightings inside the fish farm area compared to the total number of sightings, were: (i) only one individual had a percentage lower than 10% (8.3%); (ii) nine had a percentage between 10 and 30%; (iii) four had a percentage between 30 and 70%; (iv) three had a percentage between 70 and 99%; and vi) three were sighted exclusively inside the fish farm area and nowhere else (a percentage of 100%). None of these animals was seen foraging inside the fish farm area in P2. The mean number of sightings per individuals

**Table 2 Structure of the communities.** Table of the communities obtained with the Maximum Modularity method for the entire 4 population, P1 and P2 datasets. Total number of individuals (N° Ind.), percentage of "cage" (C) and "no-cage" (NC) individuals, females (F), males (M) and sex not determined (ND), MSR (mean sighting rate), SD (standard deviation).

| Community | N° Ind. | C | NC | F | M | ND | MSR ± SD |
|---|---|---|---|---|---|---|---|
| | | | Entire population ($N = 85$) | | | | |
| 1 | 30 | 13% | 87% | 43% | 10% | 47% | 0.03 ± 0.02 |
| 2 | 16 | 38% | 63% | 63% | 25% | 13% | 0.04 ± 0.02 |
| 3 | 6 | 100% | 0% | 17% | 17% | 67% | 0.04 ± 0.03 |
| 4 | 19 | 26% | 74% | 37% | 26% | 37% | 0.06 ± 0.05 |
| 5 | 14 | 29% | 71% | 71% | 0% | 29% | 0.08 ± 0.05 |
| | | | P1 ($N = 32$) | | | | |
| 1 | 11 | 73% | 27% | 73% | 9% | 9% | 0.07 ± 0.03 |
| 2 | 2 | 100% | 0% | 0% | 50% | 50% | 0.06 ± 0.01 |
| 3 | 8 | 100% | 0% | 13% | 25% | 63% | 0.13 ± 0.08 |
| 4 | 11 | 9% | 91% | 45% | 9% | 45% | 0.05 ± 0.02 |
| | | | P2 ($N = 69$) | | | | |
| 1 | 24 | 38% | 63% | 71% | 8% | 21% | 0.08 ± 0.05 |
| 2 | 27 | 11% | 89% | 33% | 15% | 52% | 0.04 ± 0.02 |
| 3 | 17 | 18% | 82% | 41% | 24% | 36% | 0.08 ± 0.06 |

was 10.1 (SD = 6.4, range 5–35), while the mean sighting rate was 0.08 (SD = 0.05). Pooled HWIs calculated for each dyad gave a mean of 0.09 ($n = 1024$, SD = 0.04; max HWI ± SD = 0.49 ± 0.19; Table 3). The social differentiation of 0.957 indicated a society with good differentiation and the correlation between true and estimated association indices of 0.628 indicated a reasonably useful representation of the social structure (*Whitehead, 2008b*; *Ansmann et al., 2012*; Wiszniewski et al., 2009).

The P2 dataset included 69 individuals: 16 C and 53 NC; 34 females, 11 males and 24 of unknown sex. Among these 69 individuals, 21 were also sighted in P1. C individuals were never seen foraging in the fish farm area in P2. The mean number of sightings per individuals was 21.5 (SD = 17.8, range 1–75), while the mean sighting rate was 0.06 (SD = 0.05). In P2, pooled HWIs calculated for each dyad ($n = 4761$) gave a mean of 0.07 (SD = 0.02; max HWI ± SD = 0.37 ± 0.11; Table 3). The estimate of social differentiation was 0.959, which indicated a well differentiated population (*Whitehead, 2008b*). The correlation between true and estimated association indices (0.697) indicated a reasonably good representation of the social system (*Ansmann et al., 2012*; Wiszniewski et al., 2009). The segregation between C and NC individuals in P1 (Mantel test) revealed that individuals were associated preferentially with those belonging to the same foraging strategy (HWI within = 0.15 ± 0.07, HWI between = 0.04 ± 0.04; $t = 8.281$, $p < 0.0001$), while in P2 no significant differences were found (HWI within = 0.07 ± 0.04, HWI between = 0.07 ± 0.02; $t = 0.074$, $p = 0.9410$).

In P1 different communities were detected by cluster analysis and the CCC value of 0.83 was high enough to ensure a good representation of the association matrix. The MM of the population detected a division of the society (MM = 0.406). Community 1 was composed

Frau et al. (2021), *PeerJ*, DOI 10.7717/peerj.10960

**Table 3 Social network metrics and HWIs.** (a) Social network metrics of the entire population and comparison between cage (C) and no cage (NC) individuals in P1, cage (C) and no cage (NC) individuals in P2 and P1 and P2 datasets. P value from two sample permutation test. Significant results in bold. (±SD). (b) Mean and Max HWIs of the entire population and comparison between P1 and P2 and cage (C) and no cage (NC).

| (a) | Entire population | P1 | | | P2 | | | P1 | P2 | *P value* |
|---|---|---|---|---|---|---|---|---|---|---|
| | | C | NC | *P value* | C | NC | *P value* | | | |
| **Strength** | 4.24 (1.75) | 2.91 (1.16) | 2.81 (1.06) | *p* = 0.8067 | 4.83 (1.59) | 4.59 (1.57) | *p* = 0.6091 | 2.87 (1.10) | 4.65 (1.56) | ***p* < *0.0001*** |
| **Reach** | 21.03 (10.13) | 9.33 (3.81) | 9.49 (3.91) | *p* = 0.9106 | 25.56 (9.40) | 23.55 (9.55) | *p* = 0.4731 | 9.40 (3.79) | 23.98 (9.49) | ***p* < *0.0001*** |
| **Clust. Coeff.** | 0.15 (0.05) | 0.20 (0.07) | 0.26 (0.07) | *p* = 0.0446 | 0.19 (0.04) | 0.20 (0.05) | *p* = 0.1377 | 0.23 (0.07) | 0.20 (0.05) | *p* = 0.0871 |
| **Affinity** | 4.82 (0.58) | 3.20 (0.20) | 3.33 (0.32) | *p* = 0.2116 | 5.24 (0.37) | 5.05 (0.50) | *p* = 0.1014 | 3.25 (0.26) | 5.09 (0.48) | ***p* < *0.0001*** |
| **(b)** | Entire population | P1 | | | P2 | | | C | NC | |
| **HWI mean** | 0.05 (0.02) | 0.09 (0.04) | | | 0.07 (0.02) | | | 0.06 (0.04) | 0.06 (0.02) | |
| **HWI max** | 0.36 (0.13) | 0.49 (0.19) | | | 0.37 (0.11) | | | 0.30 (0.18) | 0.36 (0.11) | |

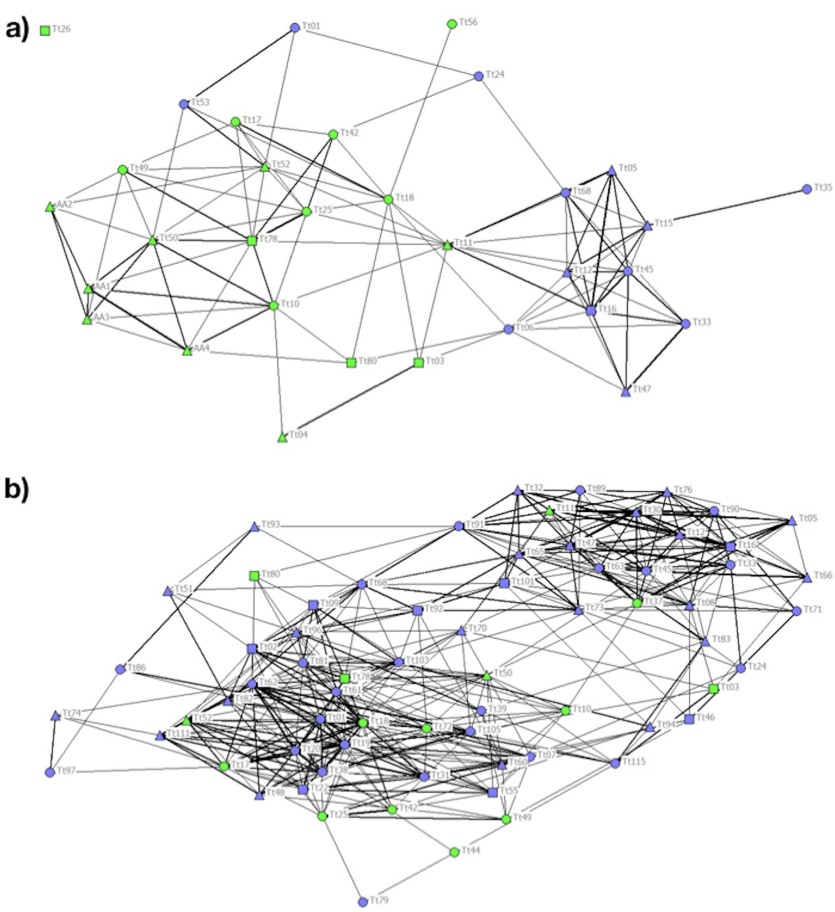

**Figure 3** **Social network representation of the (a) Period 1 (b) and Period 2 population, showing only the significant associations from the permutation test.** Individuals are identified by their ID code; thickness of lines is proportional to HWI values. Nodes shapes represent different sexes (triangle = ND, circle = females, square = males). Green nodes are cage individuals (C), blue nodes are no cage individuals (NC).

mainly of females and C individuals. Communities 2 and 3 were composed exclusively of C individuals, while Community 4 was composed mainly of NC individuals (Table 2). The stronger associations between individuals were those between animals adopting the same foraging strategy (Fig. 3A).

In P2 the cophenetic correlation coefficient did not ensure a good representation of the association matrix (CCC = 0.763). Nonetheless, the MM of Newman's eigenvector method (MM = 0.315) supported the division of the population into three clusters (Table 2). The majority of the 20 C individuals clustering together with other individuals adopting the same feeding strategy in P1 spread out over the network in P2, regardless of foraging strategy (Fig. 3B).

In P1, two-sample permutation tests carried out for each network measure between C and NC individuals revealed a significant difference for the clustering coefficient only, while no differences were detected in P2 (Table 3). The comparison of the network measures

between P1 and P2 revealed higher strength, reach and affinity in P2 compared to in P1, indicating a more connected network, while no difference was found in the clustering coefficient (Table 3).

## DISCUSSION

### Social structure of the entire population

The social structure of bottlenose dolphins in Alghero was well differentiated, with preferred and avoided long-lasting relationships between certain individuals and a probable clustering into five communities; two of them featured the strongest bonds. Unlike in studies of other bottlenose dolphin populations (*Smolker et al., 1992*; *Morteo, Rocha-Olivares & Abarca-Arenas, 2014*; *Moreno & Acevedo-Gutierrez, 2016*), we did not find sex segregation, but the sex-specific social structure highlighted some interesting aspects: some females are more connected with each other and form stronger and long-lasting associations with a subset of females, similarly to *Connor et al. (2000)* and *Mann (2000)*. This may suggest that living in a highly connected network leads to some advantages for females, mainly in terms of survival and protection of the calves from predators, as well as defence from harassing males (*Mann, 2000*). This pattern was also detected in other Mediterranean sites. For example, in the Aeolian Archipelago (Italy) females in similar reproductive status formed strong association (*Blasi & Boitani, 2014*), while in the Straits of Sicily social relationships among post-parturition females were identified (*Papale et al., 2016*).

Some studies reported that male bottlenose dolphins can build strong and long-lasting relationships with one or two individuals of the same sex, called alliances (*Wells, 1991*; *Connor, Smolker & Richards, 1992*; *Krützen et al., 2004*; *Moreno & Acevedo-Gutierrez, 2016*). Although different factors can impact alliance formation (e.g., prey and predator abundance and habitat use; *Connor & Krützen, 2015*; *Rako-Gospic et al., 2017*), increased male–male competition seems to be the best predictor of alliance formation (*Ermak, Brightwell & Gibson, 2017*), facilitating female acquisition and defence against other alliances (*Smolker et al., 1992*; *Connor et al., 2011*). Males in the Gulf of Alghero seem to build only short-term and weak relationships. The absence of strong male alliances and/or a great variability in their association pattern was observed in some other areas and the Mediterranean Sea (*Wilson, 1995*; *Blasi & Boitani, 2014*; *Genov et al., 2019*). The lack of strong bonding between males was associated with the low density of dolphins in the area, with a low encounter rate between individuals (*Connor & Whitehead, 2005*), which can lead to little intra-sexual competition. In our case, due to the low number of certain males ($n = 13$) compared to non-sexed individuals ($n = 32$), any result must be taken with caution, since stronger male–male relationships could be present outside the pool of sexed individuals. An increased number of sexed individuals would allow improved knowledge about male–female relationships in the Alghero dolphin population.

### Fish farm effect on dolphin social structure

The substantial reduction in the activity of the fish farm since 2015 gave us the opportunity to compare social structure with more and less anthropogenic food patches available and to investigate the effect of the fish farm on the population's social structure. Although

the cluster analysis and sociogram did not indicate a clear separation of individuals into communities related to a different foraging strategy class, the Mantel test suggested a significant difference in association patterns between C and NC individuals in the P1 network. Conversely, in the P2 network 43 out of 69 individuals showed preferential associations, but no difference was found in the tendency to associate between individuals of the same foraging strategy class. Therefore, fish farm activity seems to be a pivotal factor affecting the social structure of this population, favouring the association of individuals engaged in the same foraging strategy. This result corroborates the evidence of preferred associations between individuals adopting similar foraging strategies already provided in other Mediterranean areas (*Díaz López & Shirai, 2008*; *Pace, Pulcini & Triossi, 2012*; *Genov et al., 2019*) consistent with the homophily principle that some animals prefer to affiliate with individuals behaving in the same way (*McPherson, Smith-Lovin & Cook, 2001*). Other than that, network metrics (Strength, Clustering Coefficient, Reach and Affinity) were all significantly higher in the P2 network than in P1. The lower values of the network measures in P1 could be the result of a lower tendency to form strong bonds when a food source is easily predictable (*Díaz López & Shirai, 2008*), as in the case of foraging inside the fish farm. Conversely, the result obtained for the P2 network could indicate the merging of individuals of the two foraging strategy classes into a more well-connected network. Thus, it seems that when the fish farm activity decreased, the social structure of the population changed, moving towards a less complex structure. As a consequence of the loss of predictable artificial food source, individuals faced a more irregular distribution of prey, and they may have needed to travel more widely to find other resources (*Gowans, Würsig & Karczmarski, 2007*; *Ansmann et al., 2012*), changing their association patterns. Even if the increased social network measured in P2 could also have been influenced by the larger number of individuals (and therefore associations), this result is coherent with the result of the Mantel test and is also consistent with the findings regarding another dolphin population that showed a similar change in the social structure after a reduction of trawler fishing activity (*Ansmann et al., 2012*).

This study did not allow identification of the mechanisms driving the social changes due to the variation in farming activity. In fact, whether the interactions with the fish farm affected the fitness of C individuals by improving energetic budget or making individuals less adaptable to environmental changes is not derivable (*Genov et al., 2019*). Furthermore, whether other factors besides the fish farm may have contributed to the variation in social structure cannot be identified. Other authors (*Genov et al., 2019*) have found a separation between two bottlenose dolphin clusters in the Gulf of Trieste (northern Adriatic Sea): one often interacting with trawlers and the other not. However, this separation seemed to be related to temporal factors rather than to the spatial segregation of the two clusters, supporting the hypothesis that fishery is only one of the drivers affecting social structure.

Only a few animals (four) left the area during P2; three of them were always sighted inside the fish farm and nowhere else in P1. Apart from these few animals, this study evidenced the general high adaptability of the bottlenose dolphin, through reorganisation of the population social structure, to respond to the change in prey availability (*Genov et al., 2019*). Thus, the social structure of bottlenose dolphins should be seen as a resilient

system capable of adapting to perturbations. Further studies will be necessary to estimate the predictability of this resilience and whether it is related to different contexts and human activities.

## ACKNOWLEDGEMENTS

We would like to thank MARETERRA GROUP for logistic support during this study. Special thanks to all the students and volunteers that help in the field and lab work.

### Funding
The authors received no funding for this work.

### Competing Interests
The authors declare there are no competing interests.

### Author Contributions
- Serena Frau conceived and designed the experiments, performed the experiments, analyzed the data, prepared figures and/or tables, and approved the final draft.
- Fabio Ronchetti, Francesco Perretti and Alberto Addis performed the experiments, analyzed the data, prepared figures and/or tables, and approved the final draft.
- Giulia Ceccherelli conceived and designed the experiments, authored or reviewed drafts of the paper, and approved the final draft.
- Gabriella La Manna conceived and designed the experiments, performed the experiments, analyzed the data, prepared figures and/or tables, authored or reviewed drafts of the paper, and approved the final draft.

### Field Study Permissions
The following information was supplied relating to field study approvals (i.e., approving body and any reference numbers):

MPA Capo Caccia Isola Piana approved a permit to observe dolphins inside its boundaries (N° 0021343/2018).

### Data Availability
Raw data are available in the Supplemental Files.

### Supplemental Information
Supplemental information for this article can be found online at http://dx.doi.org/10.7717/peerj.10960#supplemental-information.

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
