# Peer review of "The influence of fish farm activity on the social structure of the common bottlenose dolphin in Sardinia (Italy)"

_PeerJ, doi:10.7717/peerj.10960_

## Round 0.1 · original submission · Major Revisions

I am pleased to have received two very thorough and constructive reviews of your paper. But see merit in the work but also not both the science and presentation need major work to do justice to the fieldwork.

·

Basic reporting

The authors provide information on the social structure of common bottlenose dolphins in north-western Sardinia. Specifically, the authors looked at how the social network of the local dolphin population may have been influenced by (or correlated with) the presence of a fish farm, by comparing two periods of differing intensity of fish farm operation, somewhat mirroring the study of Ansmann et al. 2012, which showed changes in dolphin social structure after modifications of local fishery operation.

Overall I find the research question interesting and worth investigating, and the authors present some interesting data. It is also a good opportunity to investigate this, given the changes in fish farm operation. The time-frame seems reasonable with some decent sample size overall. I do think there may be some interesting patterns here, which may well be genuine. However, there are also some potential problems with the study, which should be resolved before this study is appropriate for publication. Moreover, while the study is certainly useful, I think the authors should exercise a bit more caution in their interpretations and conclusions, especially in light of some of the inconsistencies and issues with respect to data analysis and/or interpretation of the results.
I address the main issues further below.

The treatment of existing literature appears a bit sparse and incomplete in some places, or there are ideas which seem to have been raised in other papers, but are not adequately sourced. I suggest the authors try to expand a bit on the relevant literature, and better incorporate it into their manuscript. Some relevant references are already used, but they could be better placed in the context of this study.

The manuscript would benefit from revision of English in some places.

Experimental design

As noted above, I find the research question overall interesting, well defined and worth investigating. The authors do put it in a relevant theoretical context (although there is some room for improvement, see specific comments below).

There are some issues with the methodology, which is either not entirely clear in some aspects, or appears wrong. A few general issues below, and more specific comments later on.

A few main issues:

SAMPLE SIZE
The sample size for P1 is considerably smaller (about half) than for P2. Of course, I realise that the authors have no control over how often or how many animals are observed and that the opportunity of collecting more data do not exist anymore due to changes in fish farm operation. However, this sample size may present potential problems, because the observed social patterns may be an artefact of small sample size, rather than genuine patterns. This of course isn’t something that needs to (or can) be changed, but it should be better reflected in the manuscript, by carefully considering that this may bias the results and that the conclusions are not as straightforward as they may seem.
In relation to this, it would be good to know what the overlap of the same animals was in both periods. In other words, how many of the animals from P1 were still seen in P2, and how many of those in P2 were already seen in P1. This is needed in order to evaluate the hypothesis of network reorganization, and make sure it is not just an artefact of either small sample size or a different set of animals.

Related to this, some more information on survey effort between the two periods is needed to be able to evaluate the results. How much effort was carried out in P1 and in P2 overall? And how much effort was carried out specifically in the fish farm area in P1 compared to P2? This information is needed for the interpretation of other results.


METHODOLOGY
Some aspects of methodology is lacking clarity, in order to understand what was done and how.
For example, it is not clear from the description of methods how the fish farm was surveyed. Was it part of the existing transects (if so, how often would a transect cross a fish farm area), or were fish farms surveyed separately and deliberately? Was speed kept the same during fish farm surveys as it was on other parts of a survey, or was it slower, different, did the boat stop at fish farms...? More details are needed on this.

Also, some part related to social network analysis appear to be either wrong or unclear. Overall the general approaches seem sound, but there are some inconsistencies, potential confusion of terminology or issues that weaken the study. Some details (see detailed comments below) need to be clarified.

Validity of the findings

INTERPRETATIONS
One thing I am puzzled about is the fact that the authors state the network was clearly structured. However, this does not seem to be supported by the resulting social structure parameters (including social differentiation, CCC, modularity to some degree) or by Figures 2 and 3. The authors report network structuring, but this structuring is not that obvious, especially considering that the displayed ties (associations) in Figure 2 are already double the mean HWI (so they should already be only the stronger associations) – and yet there is no obvious patterning. So something seems amiss here. Moreover, the authors report a remarkably high social differentiation (S), which is much higher than in two comparable studies (Ansmann et al. 2012 and Genov et al. 2019), despite actually showing much less structuring. My guess is that there was an error, or that the authors used the wrong metric of social differentiation (see more details further down below).

I commend the authors for providing the raw data. To have a rough idea, I did a very quick analysis in SOCPROG, based on the full (EP) dataset provided by the authors as supplementary files, and got different results from what the authors report. First, the social differentiation was much lower than reported, whereas the suggested number of clusters was higher than reported (7 instead of 5). I do have to point out that I only did a very crude, “quick and dirty” run of the analysis, without spending a lot of time on it or going into much detail. So I may have missed something. However, I did spend a bit of time on it, and the results appear to differ. Perhaps it would be good for the authors to double-check and re-run their analysis, to identify potential errors or make sure there aren’t any.

In some places the authors appear to be confusing some of the metrics and terminology in social network analysis. They report CCC and correlation between true and estimated association indices as an indication of population structuring. However, these parameters are not used for that. Further, the authors refer to the correlation between true and estimated association indices (r) as an indication of a good representation of the network, but the reported value of r does not justify that. It is rather low, so there is high uncertainty in the results, but the authors are not adequately acknowledging that.

One thing that the authors do not refer to at all in the discussion, is the fact that animals are considered C (cage) even if they were seen interacting with cages only on a single occasion. While this isn’t inherently wrong, it may bias the results towards more animals being C, even if they do not overall rely on fish farms much. This should be addressed, but also providing the information on the frequency distribution of sightings for C individuals near fish farms. For example, what proportion of C animals were seen at a fish farm once, how many were seen there a few times, how many were seen many times, and how many were seen exclusively next to a fish farm and nowhere else. Such information can help put the results in a better perspective.

Discussion seems unusually short and vague. I think the authors should expand a bit more on the study, its implications, its caveats (!), and similarities/differences to other studies. The treatment of existing literature appears a bit sparse and incomplete in some places, or there are ideas which seem to have been raised in other papers, but are not adequately sourced. I suggest the authors try to expand a bit on the relevant literature, and better incorporate it into their manuscript. Some relevant references are already used, but they could be better placed in the context of this study.

Additional comments

OVERALL ASSESSMENT

I do think this is a worthwhile study that could add valuable new information to the knowledge on bottlenose dolphin social structure, as well as on the potential interplay between social structure and anthropogenic activities, but the issues of concern need to be carefully addressed before this manuscript is suitable for publication. I suggest the authors 1) re-check their analyses and results, 2) better explain parts of methodology, 3) correct the erroneous information or erroneous analysis, 4) provide additional information, which is missing but is needed, 5) address the caveats a bit better, and “tone down” the interpretations in the discussion, 6) better incorporate their findings into the existing literature context.
This way the study could be interesting and useful as a publication, without making “too grand” statements without considering the caveats.

I provide some specific details on these and other issues below, which further clarify some of the issues I raised, and which I hope will help the authors to improve their study.


SPECIFIC COMMENTS


TITLE: This is just a suggestion of course, but may I suggest the authors consider making the title more specific? The way it reads now looks like the paper is about social structure of the species generally. But this is not the case. It is a specific study looking at a specific question. Perhaps the title could better reflect that. Perhaps make the title more specific, or maybe at least add the area in question...?


ABSTRACT

L19-20: I suggest someone proficient in English revises this manuscript for language.

L22: Not mark-recapture. Please see comment about this further down below.

L30: See my comment about the use of “inside” fish farm later on below.

L37: You mean decreased, not increased complexity, right?

L38: Replace “anthropic” with “anthropogenic”.


INTRODUCTION

L42: It should likely be “leads” or “led” here. It cannot be “lead” here, as it is grammatically wrong.

L44: Some language re-structuring is needed here.

L66: Please add “common” in front of “bottlenose dolphin”, as this is the accepted English common name for this species. Use the full name here at first mention, but feel free to then use just “bottlenose dolphin” from here on. Something like “common bottlenose dolphins (Tursiops truncatus, hereafter ‘bottlenose dolphin’)”, or something along those lines may be good.

L68: Yes, some REFS would be nice. :)

L69: References needed.

L71-72: Some references needed here as well.

L74-75: While most references here are good examples of such studies, the Bearzi et al. 2009 is a review of the ecology of the species in the Mediterranean. Since it is not a study specifically looking at this topic, it seems a bit out of place here, and I would suggest removing it (possibly adding it to a few lines up instead, next to Bearzi et al. 2019 reference, where it is more relevant).

L77-80: The second part of this sentence (after “but”) is not a contradiction of the first part, so it is unclear why there is a “but” here. These are not two mutually exclusive things, they may both operate as simultaneous mechanism. Also, it is not clear why this is referred to as “contrasting results”. Please re-word accordingly.

L82-83: Please consider and include also the following:
- Piroddi, C., Bearzi, G., & Christensen, V. (2011). Marine open cage aquaculture in the eastern Mediterranean Sea: a new trophic resource for bottlenose dolphins. Marine Ecology Progress Series, 440, 255-266.
- Bonizzoni S, Furey NB, Santostasi NL, Eddy L, Valavanis VD, Bearzi G. Modelling dolphin distribution
within an Important Marine Mammal Area in Greece to support spatial management planning. Aquatic Conserv: Mar FreshwEcosyst . 2019;29:1665–1680. https://doi.org/10.1002/aqc.3148

L83-85: This is a bit confusing - the authors have been referring to effects of fish farms in most of this paragraph, but then suddenly switch to fisheries. And yet, then they switch back to fish farms, which seems to be the topic of this study. This is a bit inconsistent and incoherent, and does not flow well. Please restructure to improve clarity and flow. I suspect the authors wanted to point out to the Ansmann et al. study (on fisheries) with respect to changes (which is fine, but it just does not read well now) – if that is the case, I suggest they try re-structuring the paragraph a bit, to better reflect this intent.

L93: Mark-recapture techniques are not used for social structure studies. I understand what the authors meant, but there may be a bit of a misunderstanding of terms here. Mark-recapture techniques and photo-identification are not the same thing. Photo-identification is the process of photographically identifying animals (which can be used for various things, including social structure studies, as well as abundance studies through mark-recapture). However, mark-recapture techniques are a set of statistical methods used to infer survival rate, abundance and emigration rates, BASED ON photo-identification (or actual capturing and marking). Therefore, saying that mark-recapture techniques were used to study social structure is a bit nonsensical. Please remove “mark-recapture techniques”. Instead, you may want to write something like “applying social structure/network analysis on a 12-year photo-identification data” (or something along those lines).


MATERIALS AND METHODS

It is not clear from the description of methods how the fish farm was surveyed. Was it part of the existing transects (if so, how often would a transect cross a fish farm area), or were fish farms surveyed separately and deliberately? Was speed kept the same during fish farm surveys, or was it slower, different,...? More details are needed on this.

L105: Of course, the authors know better than I do, but just looking at the map, is this really considered north-western Sardinia? It looks more like pure WEST to me. Just a comment, I leave it up to the authors.

L107: Not sure what is meant by “graded”.

L108-109: Please include the information on what fish species is (was) being farmed.

L110: Just to clarify: the remaining two cages remained fully operational? The word “discontinuous” may be confusing me here, because I am not sure if the authors wanted to say that the fish farm functioning was discontinued (i.e. it stopped), or that it was discontinuous in the sense that it was operational on-off, or that it operated with progressive reduction in number of cages (but that the remaining two cages were still operational). I suspect the latter is the case, but perhaps re-word a bit to make it clearer.

L111: The authors provide the full species name here (common bottlenose dolphin). See my previous comment about this. So either use the full name throughout, or use the full name at the first mention but then use “bottlenose dolphin” in the rest of the manuscript.

L112: conditions

L124: This is minor, but whether handheld or fixed, both are “GPS” and both are “GPS plotters”. Also, not clear why one is specified to the brand (Garmin) but other not. So either provide full details on both, or just say “GPS” and leave it at that (doesn’t matter if it was handheld or fixed).

L127: classes

L133: Please replace the word “slices” - sound a bit wrong :) – with “genital slits” or even better “genital area”.

L136: I am not necessarily against using this reference, but for something basic such as determining sex based on presence of calves, it would be more appropriate to cite some earlier original sources (perhaps Shane, or Wells, or Wursig or even Bearzi), because this certainly wasn’t first described by Blasi and Boitani as late as 2014.

L139-140: “In the bottlenose dolphin marks are mainly distributed on the rear side of the dorsal fin.”
This is not needed here, and not relevant, so please delete.

L140: Remove “thus”.

L143: Remove “further” (there was no mention of classification yet, so no “further” is needed).

L153: “thus dataset”...?

L152-153: I suspect individual and its data could be assigned to one of the period, or to both of the periods, right?

L157: Please define what the perimeter of the fish farm is (because this can be defined in various ways, such as “inside marker buoys”, “2 m from the cages”, “between cages”...., which isn’t necessarily obvious or unambiguous).

L162: Yes, but in this case (and typically in photo-ID studies of delphinids) it is the rate of ASSOCIATION (not interaction) that we are able to observe. Please change to “association”.

L163: “Between the many possible”...? Remove.

L164: Bejder et al. 1998 did not come up with the HWI. Please cite the appropriate reference here, which is Cairns and Schwager 1987.

L167-168: Please add some references here (ideally of a study what already stated this same thing).

L169: non-independence

L177: Bejder, not Bedjer.

L180-181: I personally don’t think this null hypothesis needs to be stated (especially since these tests are now rather standard).

L188: I understand the aim here, but to an uninformed reader (one that is not versed in social network analysis), it isn’t clear what “random data sets” are here, the way it is written. I think you can delete the sentence of L186-188, because it isn’t needed, and you don’t need to explain everything underlying the analyses, because they are already standard and published in books etc. Just state the essential stuff and cite the relevant references such as Whitehead 2008 book, etc.

L189-191: Same.

L193: Just say “between sexes” or “between males and females”. No need to say “between sex” and then explain that this means males and females.

L199: Please specify which modularity you are referring to. Is it the Newman’s eigenvector method or something else...?

L204: Please be specific and refer to the exact threshold. Also, if you are basing this on other studies that did this, perhaps provide a citation.

L210: indices

L214: Just to double-check: before, the authors said animals seen MORE than 5 times were included. Now they are saying it is AT LEAST 5 times. Just double check that this was indeed intended to be the case. In other words, is this a different threshold or a wording error...?


RESULTS

Some more information on survey effort between the two periods is needed to be able to evaluate the results. How much effort was carried out in P1 and in P2 overall? And how much effort was carried out specifically in the fish farm area in P1 compared to P2? This information is needed for the interpretation of other results.

L229: 13081 nautical miles of what? Add “survey effort” if that was the intent.
Please provide some information on the distribution of survey effort across the two periods, and between fish farm area and non fish farm areas.

L231: This is now in contradiction to what was said earlier, where it was stated that animals seen more than 5 times were included. Which is it, “5 times or more” (cut-off is 5) or “more than 5 times” (cut-off is 6)? Please be precise, because these two things are not the same.

L232-240: It would also be good to know how often different C individuals were seen in the fish farm areas. In other words, the variable sighting rates of C individuals in a fish farm. This is because an animal seen there once versus an animal that is seen there all the time, are two different things, and it would be useful to know if there was substantial variation.
Moreover, it would be useful to know how often C individuals in P1 still interacted with cages during P2.

L233-234: I suggest reporting SD as SD = (not SD+-) (and later on in the results)

L237-238: By “inside”, I suspect you mean inside the perimeter, not inside cages themselves. Please clarify and re-word as appropriate.

L238: Please provide what % of sightings this is.

L238-239: You already provided the definitions of cage and no-cage, so no need to provide the explanations here again.

L242: Please replace “statistically” with “significantly”.

L243: I notice the authors used a Student’s t-test. This will only be appropriate if group size data are normally distributed (and I doubt the sample was large enough for this not being relevant, or for other assumptions of the t-test to be valid). But was this tested and if so, was it the case? Dolphin group size data typically follow a non-normal distribution (I cannot imagine group size data being normally distributed), rendering the t-test inappropriate. The authors should test this assumption of normality and state it in the manuscript. If data are non-normal, then a different, non-parametric test, should be used, in this case likely the Mann-Whitney U test.

L248-249: I am confused as to why two different SD’s are reported... Are you referring to the SD of the mean of the maximum HWI here...?

L249-251: “Estimate of social differentiation gave a value of 1.942 and the correlation between
the true and estimated association index (0.619) indicate a well differentiated population (Whitehead, 2008b)”
This is incorrect. The first part is OK, but the correlation between the true and estimated association indices do not indicate whether or not a population is well differentiated. It tells you if your data accurately describe the social system. Moreover, it is association indices (plural), not index (singular).
Also, the correlation of 0.619 indicates a “somewhat representative” pattern of relationships within a population (see Whitehead 2008 book), not a “good” one. Authors are not mentioning this fact, but they base a lot of conclusions on this. So some caution is needed here and authors should refer to this caveat. Also, please provide a measure of precision for the estimate of social differentiation.
(After looking at Figure 2, I now have my doubts about the social differentiation of 1.942. That is quite high number, indicating strong differentiation. And yet, this is not evidenced from Figure 2, from modularity, or from the fact that CCC suggests dendrogram is hardly feasible. If you look at the recent study of Genov et al. 2019 for example, there is strong evidence of structuring there (by the sociogram, by the dendrogram, the associated CCC and the modularity), and yet the S value is smaller than here. The same goes for Ansmann et al. 2012 study, which had high structuring in the first period, but the level of social differentiation was lower than the one reported here. Could there be an error here? I suggest the authors double-check if the S value is really correct. Moreover, which method did the authors use? The Poisson approximation or the maximum likelihood method? Note that the Poisson approximation tends to give you a higher number, but is more biased and less precise than the maximum likelihood method (see Whitehead 2008 book), so it should not really be used without good justification. If the authors used this, they should instead use the maximum likelihood method, but in any case, specify which method they are reporting.)

L251-252: This is methods, not results.

L252-256: I’m afraid the authors are confusing terms here. The way authors wrote it, it gives the impression that SDs tell you if associations are random or not, whereas CVs tell you if there are preferred or avoided associations or not. In other words, it seems like these things tell you different things. But this is not the case. Both SDs and CVs tell you whether or not associations are random. They basically tell you the same thing. And if they are non-random, then this means they must be preferred and/or avoided. It’s not like you first determine if they are non-random and then determine if they are preferred and avoided.
I suggest re-wording to something like: “Significantly higher SD and CV values in real data compared to random (you can provide numbers here) revealed the presence of non-random associations in the
population.”

L256: associated

L264: “few” or “a few”? There is a subtle difference. “Few” means “not many”. “A few” means “some”.

L265: were solitary – keep the tense consistent across the manuscript (generally past tense)

L264-265: Please be specific. How many were truly solitary and how many had a few weak associations? If they had some associations, then by definition they are not solitary.

L267-268: What do you mean by “mutually exclusive”? That they avoided each other completely? Or were you trying to say something else, that they only associated with one another? “Mutually exclusive association” sounds strange and is a bit confusing, because mutually exclusive implies that if there is one, there cannot be the other. Please re-word to make it clearer.

L269: Based on the terminology used so far, it is not clear to me what exactly the authors mean by “significant relationship”? Does this refer to 2 x mean HWI? Or to something else? Please be explicit (or define the terms very carefully in the Methods). Even if there isn’t an association of 2 x mean HWI, it still does not mean there is no relationship among animals.

L270: CCC stands for COPHENETIC correlation coefficient, please add the word “cophenetic”.

L270-271: This is not entirely accurate. CCC does not tell you whether or not your population is structured into units, but instead tells you how well your dendrogram (from hierarchical cluster analysis) describes your association matrix. These are not the same thing. And you are not showing any dendrogram anyway, so CCC has little meaning here. In fact, when I think about it, I think dendrogram would be useful here, because it would help assess some of the results you are reporting, because they seem to give mixed messages (see below).

L272: Please remove “significant”, because modularity is not a statistical test, it is instead a guidance showing how much confidence we can have in the results of assigning individuals into clusters. Values above 0.3 suggest “good” division among clusters, but 0.319 is barely above that.
So, I suggest the following (or similar) wording: “Nonetheless, the maximum modularity of 0.319 suggests a population division into five different communities”.
However, now that I look at Fig. 2, I don’t think you actually have good evidence of population division (already exemplified by the CCC value, which suggest that constructing a meaningful dendrogram may be challenging given the data). Your Figure 2 already includes only the “strong” (twice the mean HWI) associations, and even with this, there is clearly no division into clusters.

L276: Add a reference to Table 1 here.

L276-278: Based on Table 1, this is not true. Communities 2 and 3 have the highest number of C individuals, but communities 3 and 5 have the lowest total number of individuals.

L278: But...the sociogram does not show that at all. I don’t see anything in Figure 2 that suggests division into clusters. Perhaps it would help if the authors clarified what exactly they are basing this statement on.

L278-280: This sentence is very confusing. I now re-read this sentence 5 times, but it is still not clear to me what the authors are trying to say. What do you mean “three clusters connected by a third cluster and one less connected cluster”...? This sentence needs restructuring.

L290: Providing two SDs is really confusing. If this refers to the mean of the max HWI, then it should be better stated (although I’m not sure why this information would be relevant at all – but maybe it is).

L292: indices (plural)

L298-299: This is incorrect. This correlation has nothing to do with whether or not the population is well differentiated. Also, I reiterate here that the r value is relatively low, so there is limited confidence in the results.

L304: There is some misuse of terms here (see previous comment about this). CCC is not an indication of division per se.

L305: The authors are using different terminology here (communities, social units,...). Please be consistent throughout the manuscript and stick to the same term.

L306: Again, please remove the word “significant”.

L308-310: I am sorry, but this structuring is not all that clear. First, you are already showing only “strong” associations and there is no clear division (with all associations shown, the connectedness would be even greater). Second, keep in mind that the spatial distribution of nodes by NetDraw is somewhat random and arbitrary. The positions themselves do not necessarily imply structuring. Third, apart from connections being mediated by six C individuals, they are simultaneously being mediated by NC individuals as well.

L311-312: Same comment about CCC as previously.

L313: Same comment about the word “significant”.


DISCUSSION

Discussion seems unusually short and utilising relatively little existing literature. I think the authors should expand a bit more on the study, its implications, its caveats (!), and include a more thorough treatment of existing literature, especially comparing this to other relevant studies in the Mediterranean (but also elsewhere), such as by Diaz Lopez et al. (with respect to social structure in relation to fish farms), by Bearzi et al., Bonizzoni et al. and Pirrodi et al. (with respect to bottlenose dolphins and fish farms), and Genov et al. (with respect to social structure in relation to fisheries).

L324: Again, the population does not well differentiated to me at all, judging from Fig 2.

L326: “three of them featured strong bonds” - I could not really find this in the Results. If this is stated, it should be clear from the Results as well.

L326: “Unlike many bottlenose dolphin populations” – please provide some references.

L327-328: Please link this to some other studies (globally but especially Mediterranean) that also showed lack of sex segregation in bottlenose dolphin populations.

L329-330: I am not contradicting this, but this is not all that clear from the Results and it would help if it was. There is some hint that this may be the case from Fig 2 and 3, but it is not all that clear, and it is not really provided anywhere else in the Results. Moreover, the Results state rather similar HWIs for females and males (0.06 vs 0.05, with SDs of 0.03 and 0.02, so hardly any difference), so is there really any relevance in saying this?

L331-333: Yes, but you are also saying that not all females had such strong bonds, so these results do not logically lead to this statement, otherwise all females would do this.

L341: Similar as for females, it would be great if this was shown more clearly in the Results, otherwise it is difficult to evaluate this.

L342: The authors should expand on the existing literature a bit better than just this one study, especially since there has been recent work in the Mediterranean on this.

L344: I suggest some revision of sentence structure. You probably mean “few intra-sexual interactions”, or alternatively, “little intra-sexual competition”. “Few” and “competition” do not go together.

L344-347: This is a good point, and I was just about to make that point. :) It is good that the authors state this.

L351: Replace “anthropic” with “anthropogenic”.

L352-354: I respectfully disagree here, I don’t think there is good evidence of separation really. Also, when looking at Table 1, I do agree that most of community structure (however strong the evidence for these communities might be) seems to have some signal with respect to feeding strategy, but it is also not entirely clear cut. Note also that sample size for P1 is half of that of P2, which could lead to spurious results on its own.

L370: You probably mean lower complexity, not greater. If the network became more homogeneous, the complexity was reduced, not increased.

L371: This sentence is a bit awkward, some re-wording needed.

L373-734: Good!

L378-379: This statement sounds awfully familiar... If it is based on a statement in a previous study (which it seems), perhaps make a suitable reference to that study in relation to this point.

L379-382: Perhaps the authors could add some information on what proportion of P1/P2 animals were also seen in P2/P1. This could help assess the reorganization hypothesis.

Table 1: Caption says “MSR (mean sighting rate)”, but the actual table says “MRR”...

Table 2: Since different network metrics are provided, it would be good to also provide the mean HWI values for each of the periods and each of the “feeding strategy”.

FIGURE 1: I suggest spelling out “period 1” and “period 2” rather than just P1 and P2 (in the figure caption, the legend is fine as it is). Also, it may make sense to change the black point of the fish farm into something a bit more clearly visible. It is a bit “hidden” at the moment.

Reviewer 2 ·

Basic reporting

The authors have done a thorough job of integrating their work within the current literature. Most of the work is well cited (although there are missing references in line 68). The structure of the paper generally conforms to standard scientific writing although lines 171-172 the statement regarding the size of the dataset belongs in the results.

Unfortunately the writing is not clear, unambiguous English. It appears that English is not the first language of any of the authors and they have struggled to communicate their ideas in English. I have not noted all of the places where work is needed, however I counted in the abstract alone more than 10 revisions. I encourage the authors to reach out to English speakers in the field to help with the writing. The authors might try reaching out to the Society for Marine Mammalogy student group to see if anyone is willing to help revise the manuscript. A few minor points that might be missed by non-experts.

Mark-recapture refers to population estimation. The authors are just doing photo-id, which is a great technique for this work.

Line 133 – genital slits not slices

Experimental design

The experimental design is very well set up. The authors have carefully followed the recommendations set out by Whitehead (2008) in setting up their study. This work represents original work within the scope of the journal. A few more specific points are listed below.

HWI is the most common but it is also the most relevant method for situations where individuals were likely present in the group but not identified which is the case in this study.

Line 182 – 20,000 permutations may be sufficient to stabilize the p-value for this study – however the correct approach is to increase the number of permutations until the p-value stabilizes rather than using a standard cut off for all studies. I may have misinterpreted what the authors were saying here. This needs revising.

Line 213-214. I was very glad to see the authors restricted to individuals seen at least 5 times in each dataset. This is the correct approach and one that is not followed by many studies.

Lines 241-245. The methods behind these results are not in the methods section and should be included.

I would like to know how many surveys were conducted in P1 and P2. There are many more sightings in the P2 time frame however there were 8 years in P1 and 5 in P2. Were there more dolphins in P2 or was there more effort?

Validity of the findings

In general the conclusions are well supported by the data and subsequent analysis. As outlined below I have some concerns about the frequency of finding the animals in the fish farms, before they are identified as cage feeders as well as the acceptance that the community is divided.

Line 238-239 I would be very interested to see how frequently these animals were observed feeding inside the fish farm. How dependent on the fish farm were these animals for food. If they were only seen 1/30 times that may not indicate a strong reliance on the cages, but 15/30 would show heavy reliance.

Line 262-269. It is unclear why the authors discuss the differences between male and female associations given that in line 260-261 they just stated that there were no significant differences between or within sexes. This is interesting information but they need to justify including it if they plan to keep it in.

Lines 270-280 I am concerned that the authors disregard the low value of the CCC so lightly and then accept the division of the community based on modularity. I do not know the intricacies of accepting one type of division and not the other – but I would like to see the authors address this discrepancy more before I accept that there is division in the community, especially as the sociogram suggests 3 clusters not 5 in the modularity results. This same concern arises in lines 311-316.

Lines 300-303 As mentioned above I want to know more about how often dolphins were found in the fish farms. Finding them only once in the fish farm is not really a very strong measure of their feeding strategy. I think their results would be more convincing if they restricted the definition of cage dolphins to dolphins seen at least 2 (I would prefer more) in the cage area.

Additional comments

Overall this is a solid study that has been well conducted. There are a few methodological and interpretation issues I would like to see cleared up before publication. Unfortunately the manuscript needs major revising to become more readable in English.

---

## Round 0.2 · Minor Revisions

Both referees still find many parts of the paper needing clarification. In addition to a painstaking review of the points raised, I suggest you find or pay a fluent English speaker to proofread the paper for clarity.

·

Basic reporting

The authors have incorporated most of the requested revisions in a satisfactory manner. I point out a few remaining pending issues below (in comments to authors), but other than that, I have no remaining major concerns. Again, I congratulate the authors for undertaking this work.

Experimental design

The issues from the first manuscript version have largely been addressed.

Validity of the findings

No further comment.

Additional comments

My apologies to the authors for the delay in submitting this review of the revised manuscript, but these have been some seriously challenging times. Nevertheless, I hope the review is to the authors' satisfaction, and other than a few relatively minor pending issues (see below), I am happy to recommend this manuscript for publication.

And since it's the eve of 24th December: Happy holidays!



SOME MINOR PENDING ISSUES:

L31: Replace “lowered” with “lower”

L40: replace “were” with “are”

L91: My sentence “applying social structure/network analysis on a 12-year photo-identification data” was provided merely as a general example, but I see now that the authors took it verbatim. What I meant by “social structure/network analysis” was that authors can choose the appropriate term based on what they did and what they use in other parts of the text, but not that they should use the term “social structure/network analysis”. Instead, choose one, either “social structure analysis” or “social network analysis” – as long as it’s consistent with other parts of the manuscript.

L122: “Few” is a relative and uninformative term. Please be specific and provide the actual speed or speed range.

Referee wrote
L152-153: I suspect individual and its data could be assigned to one of the period, or to both of the periods, right?
Author answer: yes, it was.
Referee response: OK, but then (which was my point) this should be written. Now it says dolphins were assigned to one of the two periods, but in reality, they could be assigned to one, the other, or to both. This should be incorporated.

L235: Just double-checking: is the value 11000 correct? It is more than 5 times the amount of effort in P1, so it seems striking (given the number of surveys and sightings in P2 is only about double that of P1). This is just for the authors to double-check, if it’s correct then fine.

L298-299: The percentage of times – please specify what the percentage refers to: number of sightings in fish farm area, but out of a total of what? Presumably the total number of sightings for that individual, but it could also be something else, so please specify.

Referee wrote
L237-238: By “inside”, I suspect you mean inside the perimeter, not inside cages themselves. Please clarify and re-word as appropriate.
Author answer: done. We specified this aspect in the methods section.
Referee response: That’s fine, but to avoid any confusion and leave no space for doubt, I suggest adding the word “area” in such places, so saying “inside the fish farm area”.

L250: It suffices to report “p<0.001”.

L276: “Associations were never found between two male individuals”
I think I know what the aim was here, but the way it is written is extremely confusing. I had to read it a few times to clarify it in my head. I suppose the authors meant to say that one pair of males never associated between the two of them. Is this correct? But the way it is written gives the impression that there was never any association between any two males in the population. I suppose this is not the case, given what comes next in the sentence. So, I would recommend this is re-worded to make it clearer.

L276-278: The authors say “two male individuals had associations with only one male with a weak HWI (max HWI 0.28; SD=0.09), while four males did not show any significant relationship with other males.”
I’m confused: why are these two things different? Two males had weak associations… but four other males showed no significant relationship with other males. So they too only had weak associations. Why are these two things presented separately? Why not say all of them had weak associations or no associations? I may be missing something here, but it isn’t very clear from this sentence.

L282-283: I suggest a slight re-wording: “…division method based on modularity 1 (Newman, 2004) suggests a population division into five different communities.” This way you point to the best available evidence, rather than a “detection” (as it’s not really a detection).

Referee wrote
L290: Providing two SDs is really confusing. If this refers to the mean of the max HWI, then it should be better stated (although I’m not sure why this information would be relevant at all – but maybe it is).
Author answer: done. We wrote it in this way (n=1,024, SD=0.04; max ± SD=0.49, ±0.19).
Referee response: OK, but then it would be good to do the same in lines 254-255, to be consistent.

L323-325: “The high cophenetic correlation coefficient (CCC=0.83) indicated a good match between the dendrogram and the matrix of associations.”.
But there is no dendrogram, so this statement is non-sensical.

L377-380: I suggest a slight re-wording: “Although the cluster analysis and sociogram did not indicate a clear separation of individuals into communities related to the different feeding strategy class, the Mantel test suggests a significant difference in association patterns between C and NC individuals in P1 network.”

L396: Please replace “lower complex structure” with “less complex structure”.

Referee wrote
L378-379: This statement sounds awfully familiar... If it is based on a statement in a previous study (which it seems), perhaps make a suitable reference to that study in relation to this point.
Author answer: done. Sorry we had forgot it!!!
Referee response: This still does not appear to have been resolved, as the statement is still there without reference to a similar statement in a previous study.

L410: Please replace “Italy” with “northern Adriatic Sea”, because the study of Genov et al. 2019 was not limited to Italian waters.

Referee wrote
Table 2: Since different network metrics are provided, it would be good to also provide the mean HWI values for each of the periods and each of the “feeding strategy”.
Author answer: Sorry, this information were already reported in the result section and it would be redundant to add them also in a table.
Referee response: Yes, but the point was, you already use a table to report various metrics, so adding a HWI does not have any downsides. On the contrary, it has an advantage of making it easier to readily compare the differences you are reporting in the results. Especially since these form the main premise of the study results. I would agree it would be redundant if this table would not exist at all and just repeat HWI on its own. But since the table does exist, and you compare periods and feeding strategies, it only makes sense to add HWI to this comparison. So adding HWI is not redundant at all (and does not seem to be something that would take much effort or take up unnecessary space). It makes your results clearer and more informative.

Referee wrote
FIGURE 1: I suggest spelling out “period 1” and “period 2” rather than just P1 and P2 (in the figure caption, the legend is fine as it is). Also, it may make sense to change the black point of the fish farm into something a bit more clearly visible. It is a bit “hidden” at the moment.
Author answer: done, we changed the caption as suggested. About the fish farm, the black point in the figure represents the scaled size of the fish farm so we cannot make it bigger. In any case, the meaning of the black point is explained in the legend.
Referee response: To clarify, I did not say to make it bigger, I only suggested to make it more visible (possibly using a different symbol or colour). It was merely a suggestion to make it clearer. But if the authors wish to keep it as it is, that’s fine by me.

Reviewer 2 ·

Basic reporting

The writing is much clearer, but still has issues. I have pointed out some of the spots which require editing, but it would be valuable to have a second person who has a strong background in writing in English to suggest additional revisions.

Experimental design

Again I commend the authors in a well conducted study. I have identified one area of concern. It is unclear to me if the authors actually documented feeding in the fish farm area – or have assumed that this is the reason why the dolphins were in the farm area. The authors have convinced me that there is a difference in the social structure between the 2 periods but as the manuscript currently reads there is no evidence to support that is related to changes in feeding strategy. My concern rests partly on the definition of Cage and Non-Cage animals. The authors discuss that the animals were observed feeding in the Cage area but give no information on how they determined if they were feeding.

If the authors do have detailed behavioral data that can differentiate between simple presence in the fish farm area or were the dolphins congregating in this spot for foraging then the manuscript can be modified fairly simply to reflect these details. However if the authors did not carefully document feeding and have simply assumed that the animals were there to forage, then the manuscript needs to be revised to place much less emphasis on foraging changes driving the changes in social structure. I agree that it is a reasonable assumption that the dolphins congregated in the fish farm area for foraging, but they could have also congregated in this spot to reduce predation risk, or for many other reasons that the authors have not addressed. These modifications will be especially important in the discussion.

Validity of the findings

As noted above, I have begun to question the assumption that foraging changes are at the base of this change in social structure (although it certainly could be). The authors need to address why they think that foraging is the key component here (if they did not observe changes in foraging behavior).

Additional comments

Overall I commend the authors for a much improved manuscript. It is much easier to understand what the authors did.

Suggested line edits.

Line 15: change anthropized word to something different – I suggest urbanized
Line 24: delete the word relevant
Line 29: change former to earlier
Line 63-64: remove reference to Stanton and Mann 2014 as these are likely not Tursiops truncatus
Line 72: should be gear not gears
Line 75: should be prey not preys
Line 80: should be seem not seems
Line 87: consider adding the word disappearance of so it reads “likely due to the disappearance of different feeding…”
Line 93: should read “and clustering in different communities…”
Line 94: remove word has
Line 160-161: The definition of Cage and Non-cage animals is slightly confusing. If Cage animals had to be observed feeding in the farm area, and Non-cage animals were never seen inside the fish farm, were there any instances of an animal being observed in the fish farm at least once, but never observed feeding in the cage area. Also you have not given any information on how you determined that the dolphins were feeding. These definitions should be modified, and if you retain the requirement that dolphins be observed feeding – you need to include some additional information on your behavioral observations.
Line 164-165 has similar confusion. Did you simply compare the size of groups inside and outside the fish farm – or did you compare groups that were actively feeding?
Line 245-246: does this refer to presence in the fish farm – or observations of feeding behavior – if it is feeding behavior I would be interested to know how often feeding behaviors were observed outside of the fish farm as well as observations in the fish farm where the animals were not feeding.
Line 248-252: Clarify if observed feeding or just observed within the fish farm area.
Line 284: should read “similar to what was observed in the other communities.”
Line 287: the end of the sentence does not make sense and requires revision.
Line 290: should this indicate the lowest number of C individuals?
Line 302-303: Were these 3 individuals sighted in P2? If not this is very interesting – did this group of dolphins which seemed to concentrate in the fish farm leave the area when fishing activity declined? This is worth talking about in the discussion. Perhaps in Line 413.
Line 303-304: The sentence about feeding is repeated in the section below and fits better there. I would like to see here is some information about the frequency of observing foraging in P1 in the fish farm area.
Line 311-312: Were they not observed feeding in the fish farm or not observed in the fish farms at all?
Line 319-320: Need support to state that this is based on feeding strategy or distribution (which may have foraging at its core)
Line 334-335: see comments above about feeding strategy (and throughout discussion)
Line 353: should be form instead of forms
Line 354: should be social relationships
Line 376: should be available rather than availability
Line 387: should be consistent rather than consistently
Line 389-390: This sentence needs revising – I am not sure what the authors mean.
Line 396: Should be less complex structure
Line 397: should be prey not preys
Line 413: It would be helpful to state if these were all C animals.

---

## Round 0.3 · accepted · Accept

Thank you for attending to the referees' comments, largely to clarify sentences and other points of understanding.